# Loss of ERβ Disrupts Gene Regulation in Primordial and Primary Follicles

**DOI:** 10.3390/ijms25063202

**Published:** 2024-03-11

**Authors:** Eun Bee Lee, V. Praveen Chakravarthi, Ryan Mohamadi, Vinesh Dahiya, Kevin Vo, Anamika Ratri, Patrick E. Fields, Courtney A. Marsh, M. A. Karim Rumi

**Affiliations:** 1Department of Pathology and Laboratory Medicine, University of Kansas Medical Center (KUMC), Kansas City, KS 66160, USA; elee9871@gmail.com (E.B.L.); praghavulu@kumc.edu (V.P.C.); r346m406@ku.edu (R.M.); vinesh.dahiyampharm@gmail.com (V.D.); vokevin816@gmail.com (K.V.); aratri@kumc.edu (A.R.); pfields@kumc.edu (P.E.F.); 2Department of Obstetrics and Gynecology, University of Kansas Medical Center (KUMC), Kansas City, KS 66160, USA; cmarsh2@kumc.edu

**Keywords:** primordial follicle growth activation, estrogen receptor β, primordial follicles, primary follicles, transcriptome analysis

## Abstract

Loss of ERβ increases primordial follicle growth activation (PFGA), leading to premature ovarian follicle reserve depletion. We determined the expression and gene regulatory functions of ERβ in dormant primordial follicles (PdFs) and activated primary follicles (PrFs) using mouse models. PdFs and PrFs were isolated from 3-week-old *Erβ* knockout (*Erβ^null^*) mouse ovaries, and their transcriptomes were compared with those of control *Erβ^fl/fl^* mice. We observed a significant (≥2-fold change; FDR *p*-value ≤ 0.05) deregulation of approximately 5% of genes (866 out of 16,940 genes, TPM ≥ 5) in *Erβ^null^* PdFs; ~60% (521 out of 866) of the differentially expressed genes (DEGs) were upregulated, and 40% were downregulated, indicating that ERβ has both transcriptional enhancing as well as repressing roles in dormant PdFs. Such deregulation of genes may make the *Erβ^null^* PdFs more susceptible to increased PFGA. When the PdFs undergo PFGA and form PrFs, many new genes are activated. During PFGA of *Erβ^fl/fl^* follicles, we detected a differential expression of ~24% genes (4909 out of 20,743; ≥2-fold change; FDR *p*-value ≤ 0.05; TPM ≥ 5); 56% upregulated and 44% downregulated, indicating the gene enhancing and repressing roles of Erβ-activated PrFs. In contrast, we detected a differential expression of only 824 genes in *Erβ^null^* follicles during PFGA (≥2-fold change; FDR *p*-value ≤ 0.05; TPM ≥ 5). Moreover, most (~93%; 770 out of 824) of these DEGs in activated *Erβ^null^* PrFs were downregulated. Such deregulation of genes in *Erβ^null^* activated follicles may impair their inhibitory role on PFGA. Notably, in both *Erβ^null^* PdFs and PrFs, we detected a significant number of epigenetic regulators and transcription factors to be differentially expressed, which suggests that lack of ERβ either directly or indirectly deregulates the gene expression in PdFs and PrFs, leading to increased PFGA.

## 1. Introduction

The earliest step in ovarian folliculogenesis is the formation of primordial follicles (PdFs) with the breakdown of germ cell nests [1]. Two classes of PdFs are formed in mammalian ovaries, each exhibiting a distinct developmental dynamic [2,3]. While the first wave of PdFs is activated rapidly into primary follicles (PrFs) as they are formed, the second wave of PdFs mostly remains dormant and serves as an ovarian reserve throughout adult life in females [2,3]. The second wave of PdFs is selectively activated through a strictly regulated mechanism known as primordial follicle growth activation (PFGA). In the mouse, the first wave of follicles wane in the first 12 weeks of life, and then all activated follicles derive from the second wave of PdFs [2,3]. Thus, the initial quantity of second-wave PdFs, the rate of PFGA, and the loss of follicle reserve are the key determinants of female reproductive longevity. 

The mammalian ovarian reserve is represented by a fixed number of PdFs of second-wave origin that remain quiescent until recruited into the growing pool [1]. An increased rate of PFGA can lead to early depletion of the ovarian reserve, resulting in ovulatory dysfunction, including premature ovarian insufficiency (POI) [4,5,6,7]. Thus, understanding the precise molecular mechanisms that maintain PdFs in a dormant state and allow for the gradual activation of PrFs is critical and clinically important [8]. PFGA is gonadotropin-independent and involves intraovarian mechanisms [9,10,11,12]. It has been shown that secreted factors like AMH from activated ovarian follicles may act on PdFs and exert the inhibitory effect of PFGA [13,14]. Previous studies have suggested that PFGA is inhibited by gatekeepers upstream or within the PI3-kinase, mTOR, Hippo, and TGFβ signaling pathways [7,15,16]. Several transcription factors, including FOXO3A, and FOXL2, play important roles in controlling PFGA [7,15]. However, the role of estrogen signaling in PFGA was not known before our observation that estrogen receptor β (ERβ) is essential for regulating PFGA [17].

There have been contradictory reports on the role of estrogen signaling during oocyte nest breakdown and the formation of PdFs [18,19,20,21]. Aromatase knockout (ArKO) mice lacking estrogen synthesis had increased numbers of PrFs at 12 weeks of age and reduced total follicles at one year [22]. Despite these findings, it was not suspected that estrogen signaling regulates PFGA [22]. We observed that loss of ERβ did not affect the total number of ovarian follicles but markedly increased PFGA [17]. Disruption of ERβ signaling, but not ERα, resulted in excessive PFGA, leading to premature depletion of ovarian follicles [17]. Thus, ERβ plays a gatekeeping role in maintaining the ovarian reserve [17]. Targeted deletion of the ERβ DNA binding domain (DBD) increased PFGA like that of *Erβ* knockout (*Erβ^null^*) ovaries, indicating that the canonical transcriptional function of ERβ is essential for this regulation [17]. 

ERβ is a ligand-activated transcription factor that regulates cellular gene expression at the transcription level. Therefore, it is very likely that ERβ either downregulates the expression of genes that activate PFGA or upregulates the genes that inhibit this process. As the core components of PFGA are PdFs and PrFs, we primarily focused on these ovarian follicles. We investigated the transcriptome changes before, during, and after PFGA of PdFs in the absence or presence of ERβ. We isolated the PdFs and PrFs from 3-week-old *Erβ^null^* and age-matched wildtype mouse ovaries, examined the expression of ERβ mRNA and protein in isolated PdFs and PrFs, and performed RNA-sequencing analyses. Previous studies on *Erβ^null^* mice ovaries have identified genes related to steroidogenesis, preovulatory follicle maturation, and ovulation induction. In this study, we have emphasized the question of whether the loss of ERβ impacts epigenetic and transcriptional regulators in ovarian follicles. Our results indicate that ERβ is essential in upregulating the gene expression in dormant PdFs and activated PrFs. 

## 2. Results

### 2.1. Both Primordial and Primary Follicles Express ERβ mRNA and Protein

To identify the transcriptional regulatory role of ERβ in PFGA, first, we examined the expression of ERβ in mouse PdFs and PrFs at mRNA and protein levels (Figure 1 and Figure 2). We detected that *Erβ* mRNA is expressed in both PdFs and PrFs isolated from mouse ovaries (Figure 1A–C). Although the mRNA level was slightly higher in PrFs, it was not statistically significant. 

To verify further, we examined the expression of ERβ protein in isolated mouse PdFs and PrFs using immunofluorescence (IF) staining (Figure 2). Isolated PdFs and PrFs were used to prepare cytospin slides, and the follicles were stained with antibodies against total ERβ and phosphorylated ERβ (pERβ, S105). We observed that total ERβ protein is localized within the cytoplasm and nucleus of granulosa cells (GCs) as well as oocytes in both PdFs and PrFs (Figure 2A,B). In contrast, pERβ was detected only within the nuclei of GCs and oocytes (Figure 2E,F). *Erβ^null^* follicles were negative for the IF staining of total ERβ (Figure 2I,J), so we did not examine the localization of pERβ in *Erβ^null^* follicles.

### 2.2. Differential Expression of Follicular Genes in Erβ^null^ Primordial Follicles

We compared the transcriptomes of *Erβ^null^* PdFs with those of *Erβ^fl/fl^* PdFs. Of 43,230 mouse genes in the reference genome GRCm39, RNA-Seq analyses detected 21,122 genes with a TPM value ≥ 1.0 and 16,940 genes with a TPM value ≥ 5.0 in the PdFs. We observed that approximately 5% of the genes (866 out of 16,940 genes, TPM value ≥ 5) were differentially expressed in *Erβ^null^* PdFs (≥2-fold change; FDR *p*-value ≤ 0.05). Notably, about 60% (521 out of 866) of the differentially expressed genes (DEGs) were markedly upregulated, and the remaining 40% of the DEGs were downregulated, indicating that ERβ can either enhance or repress gene expression in PdFs (Figure 3A,B). The top 10 upregulated genes in the *Erβ^null^* PdFs include *Av320801*, *H2ac19*, *Or11a4*, *Gm14147*, *Gm5795*, *Gm8947*, *Gm21103*, *Gm12184*, *Pramel28* and *Or8b41*, whereas the top 10 downregulated genes are *Gm49388*, *Nutf2*, *Fam151a*, *Vsx2*, *Gm49378*, *Pabpn1l*, *H3c2*, *Tead3*, *Dnmt1*, and *Gdpd2* (Appendix A).

### 2.3. Differential Expression of Follicular Genes in Erβ^null^ Primary Follicles

We also analyzed the transcriptome profile in *Erβ^null^* PrFs and compared it with the genes expressed in the *Erβ^fl/fl^* PrFs (Figure 4A,B). Out of 43,230 genes in GRCm39, RNA-Seq analyses detected 21,356 genes with a TPM value ≥ 1.0 and 21,221 genes with a TPM value ≥ 5.0. We observed that approximately 8% of the genes (1786 out of 21,221 genes, TPM ≥ 5) were differentially expressed in the *Erβ^null^* PrFs (≥2-fold change; FDR *p*-value ≤ 0.05). In *Erβ^null^* PrFs, 83% of the DEGs (1479 out of 1786) were downregulated, whereas only 17% were upregulated, indicating that the presence of ERβ is required for upregulating the inactivated PrFs. Thus, ERβ is not only required for gene regulation in ovarian follicles before PFGA (i.e., PdFs) but also in ovarian follicles after PFGA (i.e., PrFs) (Figure 3 and Figure 4). The top 10 upregulated genes in *Erβ^null^* PrFs include *Gm5128*, *Rhox4a2*, *Gn11757*, *Or4x18*, *H2bc23*, *Gm5798*, *Gm45799*, *Mageb1*, *Gm20605* and *Vmn1r242*, whereas the top 10 downregulated genes are *Fam177a*, *Gm7903*, *H4c18*, *Zfp968*, *Gm14288*, *Ott*, *Map11c3a*, *Thoc7*, *Pigy1*, and *Derpc* (Appendix A).

### 2.4. Differential Expression of Follicular Genes during PFGA

We also identified the DEGs during PFGA of *Erβ^fl/fl^* PdFs (Figure 5A,B). A large number of new genes are activated during the PFGA, and we observed that about 24% (4909 out of 20,743) of genes with TPM value ≥ 5 were differentially expressed (≥2-fold change; FDR *p*-value < 0.05) in *Erβ^fl/fl^* PrFs compared with *Erβ^fl/fl^* PdFs. Additionally, 56% (2765 out of 4909) of the DEGs were upregulated, and 44% of DEGs were downregulated, indicating that, in the ERβ, both gene induction as well as gene repression occur during the normal PFGA process. 

In contrast, during the PFGA of *Erβ^null^* PdFs (Figure 6 A,B), we detected that a total of only 824 out of 20,268 genes with TPM value ≥ 5 were differentially expressed in *Erβ^null^* PrFs compared with *Erβ^null^* PdFs (≥2-fold change; FDR *p*-value ≤ 0.05). Of the DEGs, most of the genes (about 93%, 770 out of 824 genes) were downregulated, indicating the importance of proper gene enhancing role of ERβ during PFGA. 

When the DEGs between the two groups (PFGA in *Erβ^fl/fl^* and PFGA in *Erβ^null^* groups) were compared, we detected that only 546 genes were common and the rest of the DEGs were group-specific (Figure 7A). We observed that 4363 genes that were differentially expressed in *Erβ^fl/fl^* follicles during PFGA were missing in *Erβ^null^* follicles during PFGA. Instead, 278 ERβ-independent genes were differentially expressed in *Erβ^null^* follicles during their PFGA (Figure 7A). 

To identify the ERβ-regulated genes that play a role in PFGA, we also compared the DEGs between *Erβ^null^* PdFs and *Erβ^fl/fl^* PdFs (866 genes; Figure 3) with the DEGs between *Erβ^null^* PrFs and *Erβ^fl/fl^* PrFs (1786 genes; Figure 4). We observed that only 168 genes were common to these two groups suggesting that 1618 genes were differentially expressed in *Erβ^null^* PrFs during PFGA (Figure 7B). These findings suggest that, while *Erβ^null^* follicles lack the genes that are expressed during the PFGA of *Erβ^fl/fl^* follicles, they nevertheless expressed a large number of aberrant genes, which may be responsible for the abnormal phenotypes of activated *Erβ^null^* follicles.

### 2.5. ERβ Regulation of Epigenetics and Transcription Factors in Primordial Follicles

When we compared the transcriptomes in *Erβ^null^* PdFs to *Erβ^fl/fl^* PdFs, *Erβ^null^* PrFs to *Erβ^fl/fl^* PrFs, and *Erβ^null^* PrFs to *Erβ^null^* PdFs, we observed a consistent deregulation of genes, which suggests that ERβ plays a crucial role in transcriptionally regulating the genes in ovarian follicles before and during PFGA. Accordingly, we further analyzed the DEGs that were identified in *Erβ^null^* PdFs for transcriptional and epigenetic regulators.

Among the 866 DEGs in *Erβ^null^* PdFs (≥2-fold change; FDR *p*-value < 0.05, TPM value ≥ 5), we identified a differential expression of 26 epigenetic regulators and chromatin remodelers (Table 1). Remarkably 25 of the 26 differentially expressed epigenetic regulators were significantly downregulated in *Erβ^null^* PdFs, including *Tet3*, *Npm2*, *Mbd3*, *Ezh2*, *Dnmt1*, *Chd3 Chd4* and *Chd7* (Table 1).

We also identified 50 of the DEGs in *Erβ^null^* PdFs that were transcription factors, with 21 upregulated and 29 downregulated (Table 2). The upregulated transcription factors include *Zfp985*, *Zfp429*, *Hmx2*, *Tbx20*, *Lin28b*, *Pax5* and *Klf6*, whereas the downregulated transcription factors are *Foxl2*, *Tet3*, *Tead3*, *Pax1*, *Dnmt1*, *E2f1*, *Kmt2b*, *Mbd3*, *Fou5f1*, *Lin28a*, *Vax2*, and *E2f4* (Table 2). 

### 2.6. ERβ Regulation of Epigenetics and Transcription Factors in Primary Follicles

We further analyzed the DEGs identified in the *Erβ^null^* PrFs. Among the 1786 DEGs in *Erβ^null^* PrFs (≥2-fold change; FDR *p*-value < 0.05, TPM ≥ 5), we identified the differential expression of 97 epigenetic regulators, with 95 downregulated and 2 upregulated (Table 3). The downregulated epigenetic regulators include *Tet3*, *Pcna*, *Chd4*, *Sin3a*, *Sin3b*, *Ezh2*, *Kdm1a*, *Kdm1b*, *Gatad2a*, *Smarca2*, *Npm2*, *Prmt1*, *Setd1a*, *Dppa3*, and *Dnmt1* (Table 3).

We also detected 79 transcription factors among the DEGs, with 17 upregulated and 62 downregulated (Table 4). Important upregulated transcription factors include *Nkx6*, *Hoxb5*, *Vsx1*, *Dbx2*, and *Pou2af1*. The downregulated transcription factors include *Epas1*, *Nr5a2*, *Lhx8*, *Nobox*, *Foxl2*, *Dnmt1*, *Wt1*, *Tet3*, *Myc*, *Sox4*, *Gata4*, *Hif1a*, *Ybx2*, *Ybx3*, *E2f1*, *E2f5*, *Mbd3*, *Jund*, *Jun*, *JunB*, and *Fos* (Table 4). Among the downregulated transcription factors, the crucial roles of *Foxl2*, *Lhx8*, *Nobox*, *Nr5a2* and *Gata4* in regulating PFGA are already known [23,24,25,26,27,28]. 

## 3. Discussion

Expression of ERβ has been detected in the developing oocytes, GCs, and stromal cells surrounding the follicles, and the level of expression changes as the follicles develop [29,30,31,32,33,34,35]. While several studies have shown prominent expression of ERβ in PdFs [29,32,33], others have failed to detect expression [36]. A lack of antibody specificity has contributed to these challenges in ERβ research [34]. We observed that *Erβ* mRNA and protein are abundantly expressed in PdFs and PrFs isolated from 3-week-old mouse ovaries. Nuclear localization of phospho-ERβ indicates the presence of transcriptionally active ERβ both in the oocytes and GCs of the PdFs and PrFs. Therefore, it is expected that one should observe deregulation of gene expression following the loss of ERβ in ovarian follicles. Despite the apparent dormant state of PdFs, we observed deregulation of many abundantly expressed genes in *Erβ^null^* follicles. 

Studies have shown that somatic cells initiate PFGA by awakening the dormant oocytes [37], while signaling molecules in oocytes play a crucial role in regulating PFGA [15,38,39]. It has been suggested that signaling from activated follicles inhibits the activation of PdFs [40,41,42]. However, signaling from PdFs also inhibits the activation of neighboring PdFs [43]. These findings highlight the complexity surrounding the events leading to PFGA and the current knowledge gaps. As ERβ is expressed in both GCs and oocytes of PdFs and PrFs, disruption of ERβ signaling may impact ovarian biology, reproduction functions, and women’s health.

We observed that loss of ERβ predominantly downregulated the expression of genes both in PdFs and PrFs. This observation indicates that ERβ plays a crucial role in regulating gene expression in dormant and activated ovarian follicles. This was more clearly evident during PFGA of *Erβ^fl/fl^* and *Erβ^null^* ovarian follicles. While there was no difference in the total number of genes detected by RNA-Seq (20,743 vs. 21,221, TPM ≥ 5), there was a vast difference in gene upregulation among them (2765 vs. 307; FDR *p* value ≤ 0.05). 

ERβ is the major nuclear receptor that mediates estrogen signaling in the mammalian ovaries. Loss of ERβ can directly impair gene regulation. We observed that many epigenetic and transcription regulators are also differentially expressed following the loss of ERβ (Table 1, Table 2, Table 3 and Table 4). Expression of those epigenetic and transcriptional regulators in ovarian follicles may be regulated by the transcription function of ERβ. Thus, in addition to the direct impact of ERβ, the differentially expressed transcriptional regulators may also deregulate gene expression in *Erβ^null^* PdFs or PrFs. We observed that loss of ERβ increases PFGA and thus leads to premature depletion of PdF reserve [17]. As ERβ is a transcription factor, it is expected that this transcriptional regulator either increases the expression of genes that inhibit PFGA or decreases the expression of genes that induce PFGA.

In this study, we made a novel observation that loss of ERβ deregulates genes in *Erβ^null^* PdFs, including epigenetic and transcriptional regulators (Table 1 and Table 2). Our results suggest that such deregulation may lead to the increased susceptibility of PdFs to undergo PFGA. Moreover, following the PFGA, *Erβ^null^* PrFs also suffers from the defective expression of many genes, including many epigenetic and transcriptional regulators (Table 3 and Table 4). Such a deregulation of genes in the activated follicles ultimately leads to increased atresia, lack of follicle maturation beyond the antral stage and failure of ovulation [17,44]. Future studies are required to elucidate the underlying molecular mechanisms.

## 4. Materials and Methods

### 4.1. Animal Models

An *Erβ* mutant mouse model carrying a floxed exon 3 allele (*Erβ^fl/fl^*) [45] was included in this study. A mouse line carrying CMV-Cre [46] (006054, Jax Mice) was mated with the *Erβ^fl/fl^* mice for deletion of the floxed exon three and established heterozygous mouse lines. *Erβ^fl/null^* male and female mice were mated to generate the *Erβ^null^* mutant females. The mouse lines were maintained in C57BL/6J (000664, Jax Mice) genetic background. In all experiments, *Erβ^fl/fl^* mice were used as normal control. Three-week-old *Erβ^null^* and age-matched *Erβ^fl/fl^* female mice were euthanized to collect their ovaries and isolate the ovarian follicles. All procedures were performed following the protocols (KUMC ACUP# 2021-2601, 1/19/2022) and approved by the University of Kansas Medical Center Animal Care and Use Committee.

### 4.2. Isolation of Ovarian Follicles

Following our previously published procedure, ovarian follicles were isolated from 3-week-old mouse ovaries [17]. Approximately 100 mg of minced ovary tissue was digested in 1 mL of digestion medium (199 media containing 0.08 mg/mL of liberase with medium concentration of thermolysin (Roche Diagnostics GmbH, Mannheim, Germany) supplemented with 5 U/mL of DNase I and 1% bovine serum albumin (Thermo Fisher Scientific, Waltham, MA, USA)). The digestion mix was agitated on an orbital shaker (Disruptor Genie, Scientific Industries, Bohemia, NY, USA) at 1500 rpm for 15 min at room temperature. The enzymatic reaction was stopped by the addition of 10% fetal bovine serum. Digested ovary tissues were passed through a 70 µm cell strainer (Thermo Fisher Scientific) to remove the secondary, and large follicles and tissue aggregates. The filtrate containing the small follicles and cellular components was filtered again through a 35 µm cell strainer (BD Falcon, Franklin Lakes, NJ, USA). The 35 µm strainer was reverse eluted with medium 199 to isolate the PrFs, and the filtrate was subjected to sieving through a 10 µm cell strainer (PluriSelect USA, Gillespie Way, CA, USA) to separate the PdFs from other cellular components. Finally, the 10 µm cell strainer was reverse eluted to isolate the PdFs. Unwanted cellular components were removed from the desired follicles under microscopic examination before proceeding to RNA isolation.

### 4.3. Gene Expression Analyses in Primordial and Primary Follicles

We used 200 to 250 PdFs and 100 to 150 PrFs for cDNA synthesis using the Message Booster cDNA synthesis kit (Lucigen, Palo Alto, CA, USA). Direct cDNA and subsequent cRNA syntheses were performed by following the manufacturer’s instructions. In vitro synthesized cRNA was purified by using Monarch RNA cleanup kit (New England Biolabs, Ipswich, MA, USA) and subjected to first-strand and subsequent second-strand cDNA synthesis using the reagents provided in the Message Booster cDNA synthesis kit. The cDNA was diluted 1:10 in 10 mM Tris-HCl (pH 7.4), and 2.5 µL of the diluted cDNA was used in a 10-µL qPCR reaction as described above. The relative quantification of target mRNA expression was calculated by normalizing the data with *Actb* expression.

### 4.4. Immunofluorescence Staining of Isolated Ovarian Follicles

Isolated PdFs and PrFs were used to prepare the cytospin slides. Approximately 100 PdFs and 100 PrFs were suspended in 150 µL M199 media and loaded into a cytospin funnel, and a coated cytospin slide was placed. Then, cytospin slides were centrifuged at 700× *g* for 5 min, air-dried, and fixed in cold acetone–methanol for 10 min. Then, the slides were washed with PBST three times and blocked with 5% goat serum (Thermo Fisher Scientific) for 1 h at room temperature. The blocked slides were incubated with a rabbit monoclonal antibody against ERβ (1:250, in 5% goat serum) (Clone 68-4, Millipore Sigma, Burlington, MA, USA) or an antibody against phospho-ERβ (Ser 105) overnight at 2–8 °C. The first antibody-exposed slides were washed three times in PBST and incubated with anti-rabbit AleXa flour 594 conjugated second antibody (1:500, in 5% goat serum) at room temperature for 1 h. Slides were washed three times with PBST and covered with fluor mount with DAPI (Invitrogen), and images were captured using a Nikon-83 fluorescence microscope (Nikon Instruments, Melville, NY, USA).

### 4.5. RNA-Seq Analyses of Primordial and Primary Follicles

Gene expression at the mRNA level was evaluated by RNA sequencing (RNA-Seq). RNA-Seq libraries were prepared using the Ovation Solo RNA-Seq system (Tecan USA, Morgan Hill, CA, USA), optimized for ultra-low input RNA (10 pg to 10 ng of total RNA). Amounts of 300 to 400 PdFs and 150 to 200 PrFs were used to prepare each RNA-Seq library. Follicle lysates were used for the RNA-Seq library preparation and following the manufacturer’s instructions. The RNA-Seq libraries were evaluated for quality at the KUMC Genomics Core and then sequenced on an Illumina HiSeq X sequencer using the R1 primer provided with the kit (Psomagen, Rockville, MD, USA).

### 4.6. Detection of Differentially Expressed Genes

All RNA-Seq data have been submitted to the Sequencing Read Archive. RNA-Seq data were analyzed using CLC Genomics Workbench (Qiagen Bioinformatics, Redwood City, CA, USA) as described in our previous publications [44,47,48]. Selected RNA-Seq data were validated using the RT-qPCR analyses described above in Section 4.3.

### 4.7. Statistical Analysis

Each RNA-Seq library was prepared using the pooled follicles of three to five individual *Erβ^fl/fl^* or *Erβ^null^* mice. Each group of RNA sequencing data consisted of three different libraries. For the RT-PCR experiments, each cDNA was prepared from pooled RNA from follicles from three mice ovaries of the same genotype. Both the *Erβ^fl/fl^* and *Erβ^null^* groups consisted of >3 cDNAs. All of the laboratory investigations were repeated to insure reproducibility. The data are presented as the mean ± standard error (SE). The results were analyzed using one-way ANOVA, and the significance of the mean differences was determined by Duncan’s post hoc test, with *p* ≤ 0.05. The statistical calculations were undertaken using SPSS 22 (IBM, Armonk, NY, USA).

## Figures and Tables

**Figure 1 ijms-25-03202-f001:**
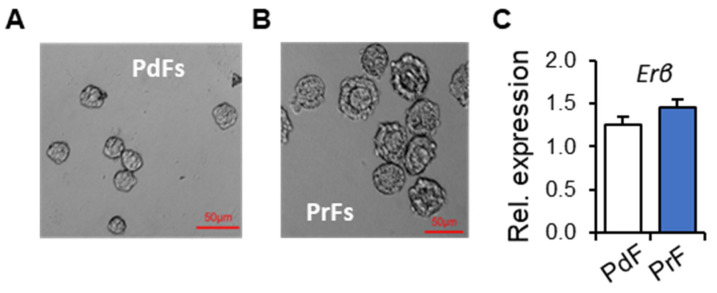
*Erβ* expression in primordial follicle (PdFs) and primary follicles (PrFs). PdFs and PrFs were isolated from 3-week-old *Erβ^fl/fl^* mouse ovaries by enzymatic digestion and size fractionation (**A**,**B**). cDNAs were prepared from the isolated PdFs and PrFs using direct ‘Cell to cDNA’ kit reagents and subjected to RT-qPCR analysis. RT-qPCR analysis shows that both PdFs and PrFs expressed *Erβ* mRNA in a comparable amount (**C**). RT-qPCR data are shown as mean ± SE, *n* ≥ 3. Rel., relative, *p* > 0.05.

**Figure 2 ijms-25-03202-f002:**
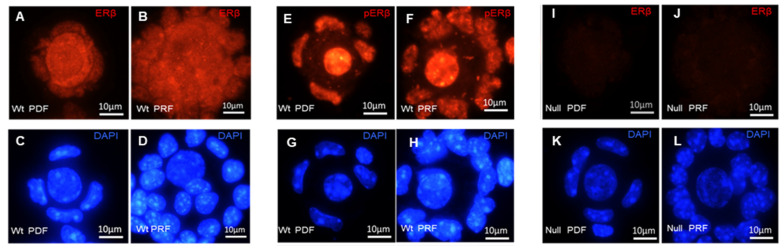
Detection of ERβ in cytospin preparations of primordial follicles (PdFs) and primary follicles (PrFs). PdFs and PrFs were isolated from 3-week-old mouse ovaries and used for the preparation of the cytospin slides. Immunofluorescence (IF) staining of the cytospin slides identified the expression of ERβ protein in both PdFs and PrFs (**A**–**L**). The upper panels show IF staining of ERβ (**A**,**B**,**E**,**F**,**I**,**J**), and the lower panels show DAPI staining (**C**,**D**,**G**,**H**,**K**,**L**). While total ERβ was detected in the nucleus and the cytoplasm of oocytes and GCs in PdF and PrF (**A**,**B**), pERβ (S105) was localized within the nuclei (**E**,**F**). *Erβ^null^* follicles were negative for ERβ detection (**I**,**J**).

**Figure 3 ijms-25-03202-f003:**
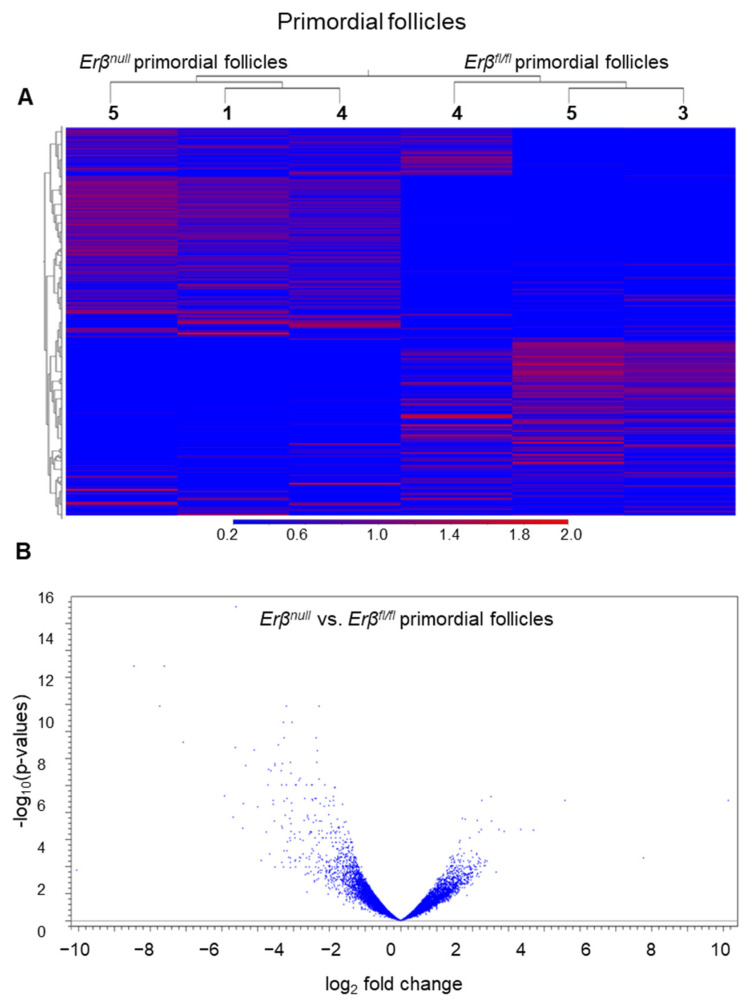
Differential expression of genes in *Erβ^null^* primordial follicle (PdFs). PdFs were isolated from 3-week-old *Erβ^null^* and age-matched *Erβ^fl/fl^* mouse ovaries. Isolated PdFs were subjected to RNA-Seq analyses. Heatmaps (all genes) (**A**) as well as volcano plots of the differentially expressed genes (DEGs) in *Erβ^null^* PdFs (**B**). In the absence of ERβ, there was an increased number of downregulated genes in *Erβ^null^* PdFs. These results also suggest that, despite the PdFs being in a dormant state, ERβ plays an important role in regulating active gene expression within them.

**Figure 4 ijms-25-03202-f004:**
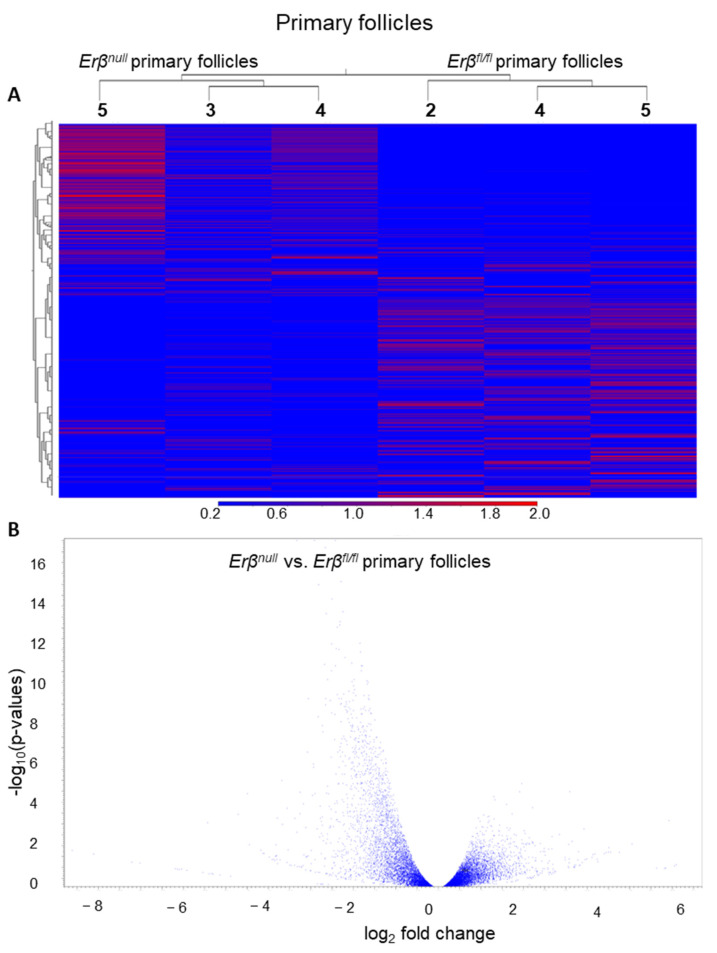
Differential expression genes in *Erβ^null^* primary follicles (PrFs). PrFs were isolated from 3-week-old *Erβ^null^* and age-matched *Erβ^fl/fl^* mouse ovaries. Isolated PrFs were subjected to RNA-Seq analyses. Heatmaps (all genes) (**A**) and volcano plots (**B**) show the differential expression of genes in *Erβ^null^* PrFs. Both heatmaps and volcano plots show that a larger number of genes are downregulated in *Erβ^null^* PrFs compared with those of *Erβ^fl/fl^* PrFs.

**Figure 5 ijms-25-03202-f005:**
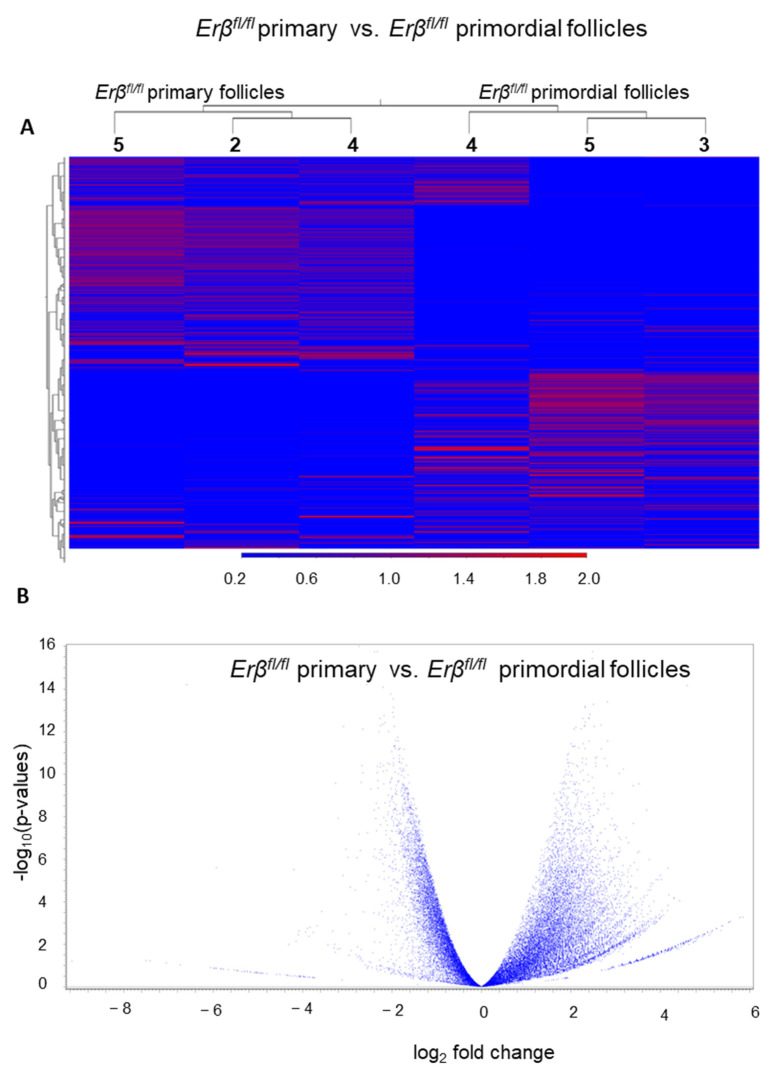
Differential expression of genes in *Erβ^fl/fl^* follicles during PFGA. PdFs and PrFs were isolated from 3-week-old *Erβ^fl/fl^* mouse ovaries. Isolated PdFs and PrFs were subjected to RNA-Seq analyses. Heatmaps (all genes) (**A**) and volcano plots (**B**) indicate the differential expression of genes in *Erβ^fl/fl^* PrFs compared with *Erβ^fl/fl^* PdFs. Both heatmaps and volcano plots show that a large number of genes are significantly upregulated during PFGA of *Erβ^fl/fl^* PdFs (**A**,**B**).

**Figure 6 ijms-25-03202-f006:**
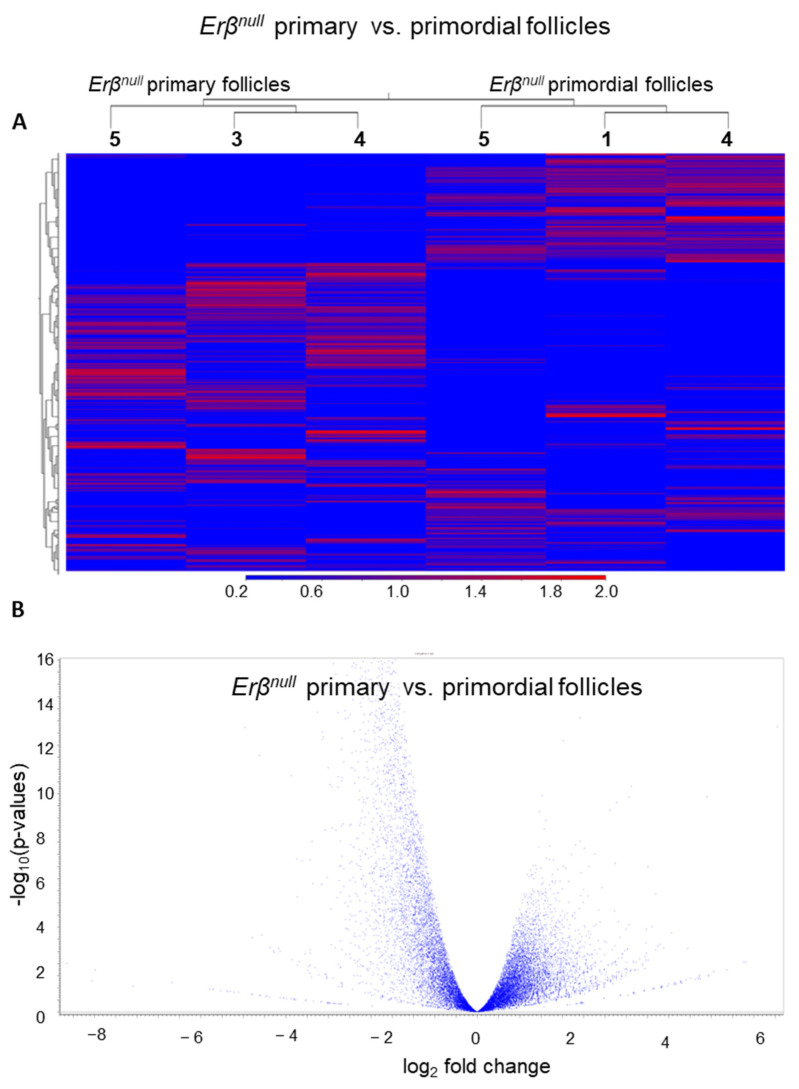
Differentially expressed genes in *Erβ^null^* follicles during PFGA. PdFs and PrFs were isolated from 3-week-old *Erβ^null^* mouse ovaries. Isolated PdFs and PrFs were subjected to RNA-Seq analyses. Heatmaps (all genes) (**A**) and volcano plots (**B**) indicate the differential expression of genes in *Erβ^null^* PRs compared with *Erβ^null^* PdFs.

**Figure 7 ijms-25-03202-f007:**
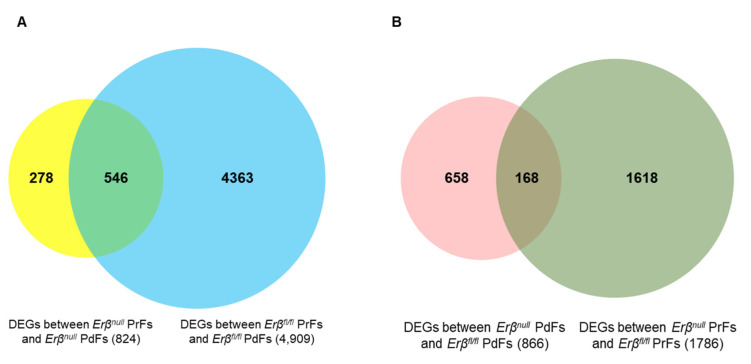
Venn diagram analysis of differential gene expression. (**A**) Venn diagram representing the differentially expressed genes (DEGs) observed during PFGA of *Erβ^null^* follicles (between *Erβ^null^* primary follicles (PrFs) and *Erβ^null^* primordial follicles (PdFs)) compared with DEGs during PFGA of *Erβ^fl/fl^* follicles (between *Erβ^fl/fl^* PrFs and *Erβ^fl/fl^* PdFs). (**B**) Venn diagram representing the DEGs between *Erβ^null^* and *Erβ^fl/fl^* PrFs compared with DEGs between *Erβ^null^* PrFs and *Erβ^fl/fl^* PrFs.

**Table 1 ijms-25-03202-t001:** Differentially expressed epigenetic regulators in *Erβ^null^* mouse primordial follicles.

Name	Chrom	ENSEMBL	Region	Max TPM	Fold Change	FDR *p*-Value
*Nek6*	2	ENSMUSG00000026749	38401655..38484618	9.22	2.38	0.04
*Chd4*	6	ENSMUSG00000063870	125072944..125107554	102.71	−2.03	0.04
*Kat6b*	14	ENSMUSG00000021767	21531502..21722546	34.22	−2.03	0.02
*Baz1b*	5	ENSMUSG00000002748	135216118..135274983	52.87	−2.04	0.02
*Ppm1g*	5	ENSMUSG00000029147	Comp (31360008..31378031)	50.81	−2.07	0.04
*Top2a*	11	ENSMUSG00000020914	Comp (98883769..98915015)	41.65	−2.17	0.02
*Chd7*	4	ENSMUSG00000041235	8690406..8867659	16.81	–2.18	0.04
*Scmh1*	4	ENSMUSG00000000085	120262478..120387383	28.73	−2.18	0.04
*Cul4a*	8	ENSMUSG00000031446	13155621..13197940	40.18	−2.20	0.01
*Chd3*	11	ENSMUSG00000018474	Comp (69234099..69260232)	33.08	−2.25	0.01
*Srcap*	7	ENSMUSG00000053877	127111155..127160391	38.99	−2.26	0.04
*Paf1*	7	ENSMUSG00000003437	28092376..28098813	43.45	−2.29	0.03
*Safb*	17	ENSMUSG00000071054	56891825..56913294	50.16	−2.46	0.03
*Idh2*	7	ENSMUSG00000030541	Comp (79744594..79765140)	45.59	−2.47	0.02
*Phf1*	17	ENSMUSG00000024193	27152026..27156882	55.85	−2.49	0.03
*Cit*	5	ENSMUSG00000029516	115983337..116147006	28.47	−2.50	0.02
*Chaf1a*	17	ENSMUSG00000002835	56347439..56379289	39.39	−2.65	0.01
*Mbd3*	10	ENSMUSG00000035478	Comp (80228373..80235384)	49.18	−2.65	0.01
*Ezh2*	6	ENSMUSG00000029687	Comp (47507073..47572275)	45.82	−2.75	0.03
*Ruvbl1*	6	ENSMUSG00000030079	88442391..88474554	28.33	−2.87	0.03
*Apex1*	14	ENSMUSG00000035960	51162425..51164596	42.76	−2.92	0.03
*Gse1*	8	ENSMUSG00000031822	120955195..121308129	19.72	−2.95	0.03
*Phf12*	11	ENSMUSG00000037791	77873580..77921365	34.51	−2.97	0.01
*Tet3*	6	ENSMUSG00000034832	Comp (83339355..83436066)	102.33	−3.49	0.02
*Npm2*	14	ENSMUSG00000047911	Comp (70884742..70896684)	274.25	−4.45	0.00
*Dnmt1*	9	ENSMUSG00000004099	Comp (20818505..20871184)	337.80	−8.16	0.00

**Table 2 ijms-25-03202-t002:** Differentially expressed transcription factors in *Erβ^null^* mouse primordial follicles.

Name	Chrom	ENSEMBL	Region	Max TPM	Fold Change	FDR *p*-Value
*Gm9048*	10	ENSMUSG00000112495	Comp (118182176..118184584)	45	7.54	0.00
*Zfp985*	4	ENSMUSG00000065999	147637734..147669655	24.54	6.53	0.00
*Zfp429*	13	ENSMUSG00000078994	Comp (67536024..67547938)	12.35	5.22	0.03
*Hmx2*	7	ENSMUSG00000050100	131150502..131159743	10.49	5.18	0.01
*Zfp988*	4	ENSMUSG00000078498	147390131..147418191	38.74	4.86	0.00
*Tigd5*	15	ENSMUSG00000103906	75781584..75786384	8.47	4.26	0.02
*Zfp595*	13	ENSMUSG00000057842	Comp (67461062..67480634)	12.72	4.19	0.03
*Bsx*	9	ENSMUSG00000054360	40785423..40791353	8.04	4.03	0.04
*Tbx20*	9	ENSMUSG00000031965	Comp (24629434..24685599)	10.79	3.79	0.00
*Zfp488*	14	ENSMUSG00000044519	Comp (33689027..33700721)	20.24	3.72	0.01
*Zfp994*	17	ENSMUSG00000096433	Comp (22416246..22444597)	12.46	3.49	0.02
*Zfp831*	2	ENSMUSG00000050600	174485327..174552625	6.34	3.10	0.03
*Lin28b*	10	ENSMUSG00000063804	Comp (45252716..45362491)	12.71	2.98	0.04
*Pax5*	4	ENSMUSG00000014030	Comp (44524757..44710487)	8.18	2.88	0.03
*Zfp850*	7	ENSMUSG00000096916	Comp (27684279..27713540)	15.1	2.67	0.02
*Hdx*	X	ENSMUSG00000034551	Comp (110479628..110606776)	10.72	2.45	0.02
*Zfp992*	4	ENSMUSG00000070605	146533480..146554749	67.95	2.44	0.03
*Klf6*	13	ENSMUSG00000000078	5911481..5920393	99.37	2.34	0.02
*Dmrta1*	4	ENSMUSG00000043753	89567673..89583009	71.39	2.29	0.01
*Csrnp3*	2	ENSMUSG00000044647	65676111..65861890	13.47	2.07	0.05
*Ikzf2*	1	ENSMUSG00000025997	Comp (69570373..69726404)	15.84	2.03	0.04
*Foxl2*	9	ENSMUSG00000050397	98837341..98840596	95.15	−2.10	0.03
*Cenpb*	2	ENSMUSG00000068267	Comp (131017102..131021987)	34.64	−2.11	0.02
*Scmh1*	4	ENSMUSG00000000085	120262478..120387383	28.73	−2.18	0.04
*Lin28a*	4	ENSMUSG00000050966	Comp (133730641..133746152)	28.17	−2.25	0.03
*Srcap*	7	ENSMUSG00000053877	127111155..127160391	38.99	−2.26	0.04
*Zfp651*	9	ENSMUSG00000013419	121588396..121600808	17.33	−2.31	0.01
*Ahdc1*	4	ENSMUSG00000037692	132738571..132805421	28.43	−2.36	0.03
*Cic*	7	ENSMUSG00000005442	24967129..24993584	41.22	−2.36	0.01
*Hsf1*	15	ENSMUSG00000022556	76361622..76386113	18.46	−2.36	0.04
*Safb2*	17	ENSMUSG00000042625	Comp (56867965..56891585)	35.58	−2.41	0.02
*Safb*	17	ENSMUSG00000071054	56891825..56913294	50.16	−2.46	0.03
*Zfp212*	6	ENSMUSG00000052763	47897410..47909573	28.34	−2.48	0.03
*E2f4*	8	ENSMUSG00000014859	106024295..106032002	47.41	−2.48	0.03
*Phf1*	17	ENSMUSG00000024193	27152026..27156882	55.85	−2.49	0.03
*Drap1*	19	ENSMUSG00000024914	Comp (5472833..5475007)	99.42	−2.52	0.03
*Pou5f1*	17	ENSMUSG00000024406	35816915..35821669	43.05	−2.61	0.03
*Mbd3*	10	ENSMUSG00000035478	Comp (80228373..80235384)	49.18	−2.65	0.01
*Kmt2b*	7	ENSMUSG00000006307	Comp (30268283..30288151)	35.67	−2.66	0.00
*Tcf7l1*	6	ENSMUSG00000055799	Comp (72603361..72766237)	17.64	−2.84	0.04
*Aebp1*	11	ENSMUSG00000020473	5811947..5822088	26.23	−3.20	0.00
*Zfp598*	17	ENSMUSG00000041130	24888661..24900990	34.89	−3.25	0.00
*Tet3*	6	ENSMUSG00000034832	Comp (83339355..83436066)	102.33	−3.49	0.02
*Sp110*	1	ENSMUSG00000070034	Comp (85504620..85526538)	134.34	−3.62	0.01
*Zfp821*	8	ENSMUSG00000031728	110432178..110451564	18.46	−3.64	0.02
*E2f1*	2	ENSMUSG00000027490	Comp (154401327..154411812)	455.5	−4.23	0.01
*Zfp414*	17	ENSMUSG00000073423	33848064..33850753	22.73	−5.25	0.04
*Dnmt1*	9	ENSMUSG00000004099	Comp (20818505..20871184)	337.8	−8.16	0.00
*Tead3*	17	ENSMUSG00000002249	Comp (28550645..28569791)	94.04	−9.96	0.00
*Vsx2*	12	ENSMUSG00000021239	84616536..84642231	159.32	−16.35	0.00

**Table 3 ijms-25-03202-t003:** Differentially expressed epigenetic regulators in *Erβ^null^* mouse primary follicles.

Name	Chrom	ENSEMBL	Region	Max TPM	Fold Change	FDR *p*-Value
*Usp44*	10	ENSMUSG00000020020	93667417..93693950	44.84	3.16	0.01
*Taf9*	13	ENSMUSG00000052293	100788087..100792568	87.70	2.37	0.03
*Anp32a*	9	ENSMUSG00000032249	62248575..62286094	97.88	−2.01	0.03
*Kmt2d*	15	ENSMUSG00000048154	Comp (98729550..98769085)	46.90	−2.02	0.00
*Sin3b*	8	ENSMUSG00000031622	73449913..73484829	33.06	−2.06	0.02
*Sf3b1*	1	ENSMUSG00000025982	Comp (55024328..55066640)	73.73	−2.08	0.00
*Ywhab*	2	ENSMUSG00000018326	163836880..163860508	100.30	−2.08	0.00
*Trrap*	5	ENSMUSG00000045482	144704542..144796588	34.95	−2.08	0.00
*Suz12*	11	ENSMUSG00000017548	79883932..79924949	97.49	−2.10	0.00
*Parp1*	1	ENSMUSG00000026496	180396489..180428819	42.15	−2.11	0.01
*Huwe1*	X	ENSMUSG00000025261	150583803..150718413	62.48	−2.11	0.00
*Sf3b3*	8	ENSMUSG00000033732	Comp (111536871..111573419)	45.78	−2.12	0.00
*Bap1*	14	ENSMUSG00000021901	30973407..30981901	34.76	−2.12	0.02
*Ogt*	X	ENSMUSG00000034160	100683666..100727957	85.85	−2.13	0.00
*Tle4*	19	ENSMUSG00000024642	Comp (14425514..14575415)	50.47	−2.14	0.00
*Crebbp*	16	ENSMUSG00000022521	Comp (3899192..4031861)	65.72	−2.17	0.00
*Noc2l*	4	ENSMUSG00000095567	156320376..156332073	27.12	−2.18	0.05
*Ncl*	1	ENSMUSG00000026234	Comp (86272441..86287122)	140.95	−2.19	0.00
*Wdr5*	2	ENSMUSG00000026917	27405169..27426547	51.40	−2.21	0.01
*Psip1*	4	ENSMUSG00000028484	Comp (83373917..83404696)	178.10	−2.23	0.00
*Mllt6*	11	ENSMUSG00000038437	97554240..97576289	25.17	−2.25	0.00
*Brd2*	17	ENSMUSG00000024335	Comp (34330997..34341608)	42.14	−2.25	0.00
*Mphosph8*	14	ENSMUSG00000079184	56905705..56934887	82.52	−2.26	0.00
*Phf1*	17	ENSMUSG00000024193	27152026..27156882	44.37	−2.26	0.01
*Max*	12	ENSMUSG00000059436	Comp (76984043..77008975)	63.02	−2.27	0.01
*Ezh2*	6	ENSMUSG00000029687	Comp (47507073..47572275)	82.70	−2.28	0.00
*Babam1*	8	ENSMUSG00000031820	71849505..71857263	41.46	−2.28	0.04
*Kdm1a*	4	ENSMUSG00000036940	Comp (136277851..136330034)	39.36	−2.30	0.01
*Ruvbl1*	6	ENSMUSG00000030079	88442391..88474554	43.20	−2.30	0.03
*Ubr5*	15	ENSMUSG00000037487	Comp (37967572..38079098)	38.86	−2.31	0.00
*Ube2b*	11	ENSMUSG00000020390	Comp (51876324..51891589)	97.87	−2.32	0.00
*Cul4b*	X	ENSMUSG00000031095	Comp (37622151..37665073)	77.46	−2.32	0.00
*Gatad2a*	8	ENSMUSG00000036180	Comp (70359726..70449034)	64.10	−2.33	0.00
*Nap1l1*	10	ENSMUSG00000058799	111309084..111334011	147.30	−2.33	0.00
*Atn1*	6	ENSMUSG00000004263	Comp (124719507..124733487)	30.78	−2.34	0.04
*Smarcc1*	9	ENSMUSG00000032481	109946776..110069246	52.40	−2.34	0.00
*Rbbp4*	4	ENSMUSG00000057236	Comp (129200893..129229163)	91.38	−2.35	0.00
*Smarca2*	19	ENSMUSG00000024921	26582450..26755722	55.11	−2.37	0.00
*Kdm1b*	13	ENSMUSG00000038080	47196975..47238755	120.51	−2.39	0.00
*Phf13*	4	ENSMUSG00000047777	Comp (152074090..152080715)	77.03	−2.40	0.00
*Ppm1g*	5	ENSMUSG00000029147	Comp (31360008..31378031)	42.81	−2.41	0.02
*Cul4a*	8	ENSMUSG00000031446	13155621..13197940	41.87	−2.42	0.00
*Smarce1*	11	ENSMUSG00000037935	Comp (99099873..99121843)	78.03	−2.46	0.00
*Pcgf6*	19	ENSMUSG00000025050	Comp (47022056..47039345)	83.94	−2.47	0.00
*Phf12*	11	ENSMUSG00000037791	77873580..77921365	48.87	−2.49	0.00
*Setd1a*	7	ENSMUSG00000042308	127375842..127399294	34.92	−2.50	0.00
*Cxxc1*	18	ENSMUSG00000024560	74349195..74354567	25.91	−2.50	0.02
*Morf4l2*	X	ENSMUSG00000031422	Comp (135633691..135644439)	100.27	−2.50	0.00
*Senp3*	11	ENSMUSG00000005204	Comp (69563941..69572910)	37.78	−2.51	0.03
*Ube2d1*	10	ENSMUSG00000019927	Comp (71090810..71121092)	47.40	−2.52	0.04
*Ddb1*	19	ENSMUSG00000024740	10582691..10607183	95.54	−2.52	0.00
*Ywhaz*	15	ENSMUSG00000022285	Comp (36771014..36797173)	255.41	−2.53	0.00
*Hdgf*	3	ENSMUSG00000004897	87813628..87823439	56.02	−2.57	0.00
*Ruvbl2*	7	ENSMUSG00000003868	Comp (45071184..45087520)	26.17	−2.59	0.03
*Tet3*	6	ENSMUSG00000034832	Comp (83339355..83436066)	110.74	−2.60	0.00
*Pcna*	2	ENSMUSG00000027342	Comp (132091082..132095234)	161.41	−2.62	0.00
*Chd4*	6	ENSMUSG00000063870	125072944..125107554	73.48	−2.64	0.00
*Hmgn1*	16	ENSMUSG00000040681	Comp (95921818..95928929)	173.80	−2.64	0.00
*Uhrf1*	17	ENSMUSG00000001228	56610321..56630486	143.72	−2.68	0.00
*Elp5*	11	ENSMUSG00000018565	Comp (69859048..69873343)	48.53	−2.69	0.03
*Hmgn2*	4	ENSMUSG00000003038	Comp (133692049..133695961)	206.22	−2.70	0.00
*Dek*	13	ENSMUSG00000021377	Comp (47238251..47259677)	100.04	−2.70	0.00
*Ube2e1*	14	ENSMUSG00000021774	4137837..4186974	84.58	−2.71	0.02
*Sfpq*	4	ENSMUSG00000028820	126915117..126930806	135.73	−2.73	0.00
*Npm2*	14	ENSMUSG00000047911	Comp (70884742..70896684)	552.61	−2.73	0.00
*Prmt1*	7	ENSMUSG00000109324	Comp (44625413..44635992)	50.43	−2.74	0.01
*Mybbp1a*	11	ENSMUSG00000040463	72332181..72342594	44.28	−2.74	0.00
*Smarcd1*	15	ENSMUSG00000023018	99600010..99611872	30.23	−2.78	0.00
*Sin3a*	9	ENSMUSG00000042557	56979324..57035650	110.49	−2.79	0.00
*Ube2t*	1	ENSMUSG00000026429	134890303..134901900	84.21	−2.83	0.01
*Mbip*	12	ENSMUSG00000021028	Comp (56375088..56392679)	49.04	−2.83	0.02
*Trim28*	7	ENSMUSG00000005566	12733041..12764962	96.42	−2.87	0.00
*Dnmt1*	9	ENSMUSG00000004099	Comp (20818505..20871184)	484.27	−2.90	0.00
*Ube2d3*	3	ENSMUSG00000078578	135143910..135173959	339.10	−2.94	0.00
*Rnf2*	1	ENSMUSG00000026484	Comp (151333755..151376706)	81.36	−2.95	0.00
*Maz*	7	ENSMUSG00000030678	Comp (126621302..126626209)	70.90	−2.95	0.00
*Nbn*	4	ENSMUSG00000028224	15957925..15992589	28.10	−2.96	0.01
*Ssrp1*	2	ENSMUSG00000027067	84867578..84877453	44.65	−2.97	0.00
*Rbbp7*	X	ENSMUSG00000031353	161543398..161562088	249.01	−2.97	0.00
*Pkm*	9	ENSMUSG00000032294	59563651..59586658	92.08	−2.98	0.00
*Exosc9*	3	ENSMUSG00000027714	36606755..36619876	35.60	−3.01	0.04
*Sap30*	8	ENSMUSG00000031609	Comp (57935741..57940894)	91.03	−3.08	0.00
*Npm1*	11	ENSMUSG00000057113	Comp (33102287..33113206)	596.28	−3.08	0.00
*Ywhae*	11	ENSMUSG00000020849	75623695..75656671	309.89	−3.09	0.00
*Clns1a*	7	ENSMUSG00000025439	97345841..97370003	63.72	−3.18	0.00
*Mta2*	19	ENSMUSG00000071646	8919239..8929667	36.75	−3.18	0.00
*Anp32b*	4	ENSMUSG00000028333	46450902..46472657	385.30	−3.23	0.00
*Skp1*	11	ENSMUSG00000036309	52122822..52137685	588.59	−3.49	0.00
*Mbd3*	10	ENSMUSG00000035478	Comp (80228373..80235384)	51.39	−3.71	0.00
*Smyd2*	1	ENSMUSG00000026603	Comp (189612689..189654560)	37.39	−3.72	0.01
*Dpy30*	17	ENSMUSG00000024067	Comp (74606469..74630939)	116.71	−3.80	0.00
*Mbd6*	10	ENSMUSG00000025409	Comp (127117825..127124887)	14.95	−3.96	0.02
*Dppa3*	6	ENSMUSG00000046323	122603369..122607231	600.49	−4.10	0.00
*Setd4*	16	ENSMUSG00000022948	Comp (93380345..93400951)	29.71	−4.18	0.01
*Sgf29*	7	ENSMUSG00000030714	126248481..126272097	44.93	−4.28	0.00
*Smarcb1*	10	ENSMUSG00000000902	Comp (75732603..75757451)	61.57	−4.78	0.00
*Actb*	5	ENSMUSG00000029580	Comp (142888870..142892509)	659.90	−4.87	0.00

**Table 4 ijms-25-03202-t004:** Differentially expressed transcription factors in *Erβ^null^* mouse primary follicles.

Name	Chrom	ENSEMBL	Region	Max TPM	Fold Change	FDR *p*-Value
*Gm28230*	2	ENSMUSG00000100642	74557072..74578262	14.34	11.27	0.04
*Batf3*	1	ENSMUSG00000026630	190830044..190841142	48.12	7.92	0.03
*Zbtb9*	17	ENSMUSG00000079605	27192141..27227350	35.39	5.54	0.02
*Barhl1*	2	ENSMUSG00000026805	Comp (28797691..28806680)	31.09	3.22	0.01
*Zfp786*	6	ENSMUSG00000051499	Comp (47796200..47807801)	44.70	3.08	0.01
*Nkx6-2*	7	ENSMUSG00000041309	Comp (139159292..139162713)	35.37	2.94	0.04
*Nkx6-3*	8	ENSMUSG00000063672	23643285..23648964	29.95	2.94	0.04
*Hoxb5*	11	ENSMUSG00000038700	96194162..96196947	47.52	2.70	0.03
*Nkx6-1*	5	ENSMUSG00000035187	Comp (101806005..101812862)	35.86	2.63	0.02
*Rax*	18	ENSMUSG00000024518	Comp (66061348..66072858)	41.42	2.60	0.01
*Vsx1*	2	ENSMUSG00000033080	Comp (150522622..150531280)	43.97	2.40	0.01
*Zscan20*	4	ENSMUSG00000061894	Comp (128477332..128503891)	49.80	2.29	0.00
*Msantd1*	5	ENSMUSG00000051246	35065356..35081183	42.68	2.26	0.04
*Dbx2*	15	ENSMUSG00000045608	Comp (95521444..95553841)	47.73	2.26	0.02
*Pou2af1*	9	ENSMUSG00000032053	51125008..51151380	75.81	2.16	0.02
*Zfp474*	18	ENSMUSG00000046886	52748987..52772902	57.25	2.15	0.05
*Zfp853*	5	ENSMUSG00000093910	Comp (143272793..143279378)	61.54	2.02	0.02
*Atf4*	15	ENSMUSG00000042406	80139385..80141742	104.74	−2.02	0.01
*Foxm1*	6	ENSMUSG00000001517	128339930..128353109	37.06	−2.04	0.01
*Esr2*	12	ENSMUSG00000021055	Comp (76167193..76224033)	49.73	−2.04	0.01
*Lhx8*	3	ENSMUSG00000096225	Comp (154011931..154036296)	50.13	−2.05	0.04
*Klf11*	12	ENSMUSG00000020653	24701273..24712788	55.50	−2.06	0.00
*Kmt2b*	7	ENSMUSG00000006307	Comp (30268283..30288151)	27.28	−2.10	0.01
*Tsc22d1*	14	ENSMUSG00000022010	76652401..76745205	92.41	−2.16	0.00
*Epas1*	17	ENSMUSG00000024140	87061128..87140838	114.54	−2.16	0.01
*Nr5a2*	1	ENSMUSG00000026398	Comp (136770309..136888186)	93.18	−2.18	0.00
*Thra*	11	ENSMUSG00000058756	98631464..98659832	48.00	−2.18	0.03
*Zfp277*	12	ENSMUSG00000055917	Comp (40365045..40495901)	44.83	−2.18	0.01
*Plagl1*	10	ENSMUSG00000019817	12936248..13007438	103.20	−2.20	0.00
*Hif1a*	12	ENSMUSG00000021109	73948149..73994304	105.08	−2.21	0.00
*Nr4a1*	15	ENSMUSG00000023034	101152150..101172676	64.97	−2.22	0.04
*Nfyc*	4	ENSMUSG00000032897	Comp (120614635..120688769)	60.97	−2.23	0.03
*Sp110*	1	ENSMUSG00000070034	Comp (85504620..85526538)	477.64	−2.24	0.01
*Nfic*	10	ENSMUSG00000055053	Comp (81232020..81291469)	26.92	−2.24	0.01
*Phf1*	17	ENSMUSG00000024193	27152026..27156882	44.37	−2.26	0.01
*Gtf2i*	5	ENSMUSG00000060261	Comp (134266688..134343614)	37.57	−2.27	0.00
*Max*	12	ENSMUSG00000059436	Comp (76984043..77008975)	63.02	−2.27	0.01
*Noto*	6	ENSMUSG00000068302	85400868..85405859	63.77	−2.29	0.04
*Gatad2a*	8	ENSMUSG00000036180	Comp (70359726..70449034)	64.10	−2.33	0.00
*Foxp4*	17	ENSMUSG00000023991	Comp (48178058..48235570)	31.10	−2.33	0.04
*Ski*	4	ENSMUSG00000029050	Comp (155238532..155307049)	116.77	−2.37	0.00
*Fbxl19*	7	ENSMUSG00000030811	127343715..127368655	31.37	−2.38	0.00
*Sox4*	13	ENSMUSG00000076431	Comp (29132902..29137696)	105.03	−2.40	0.00
*Myc*	15	ENSMUSG00000022346	61857240..61862223	61.30	−2.42	0.00
*Fosb*	7	ENSMUSG00000003545	Comp (19036621..19043976)	129.20	−2.45	0.02
*Zfp57*	17	ENSMUSG00000036036	37312055..37321527	167.57	−2.47	0.00
*Tgif1*	17	ENSMUSG00000047407	Comp (71151200..71160541)	69.51	−2.47	0.00
*Pcgf6*	19	ENSMUSG00000025050	Comp (47022056..47039345)	83.94	−2.47	0.00
*Cxxc1*	18	ENSMUSG00000024560	74349195..74354567	25.91	−2.50	0.02
*Srf*	17	ENSMUSG00000015605	Comp (46859255..46867101)	40.70	−2.51	0.03
*Gata4*	14	ENSMUSG00000021944	Comp (63436371..63509141)	72.01	−2.52	0.00
*Tcf3*	10	ENSMUSG00000020167	Comp (80245348..80269481)	48.30	−2.56	0.00
*Tet3*	6	ENSMUSG00000034832	Comp (83339355..83436066)	110.74	−2.60	0.00
*Tcf7*	11	ENSMUSG00000000782	Comp (52143198..52174158)	41.81	−2.60	0.01
*Akap8l*	17	ENSMUSG00000002625	Comp (32540398..32569581)	23.89	−2.61	0.04
*Nacc2*	2	ENSMUSG00000026932	Comp (25945547..26013232)	206.00	−2.71	0.01
*Ybx3*	6	ENSMUSG00000030189	Comp (131341818..131365439)	166.05	−2.80	0.00
*Foxl2*	9	ENSMUSG00000050397	98837341..98840596	99.70	−2.87	0.00
*Dnmt1*	9	ENSMUSG00000004099	Comp (20818505..20871184)	484.27	−2.90	0.00
*Wt1*	2	ENSMUSG00000016458	104956874..105003961	99.07	−2.94	0.00
*Gpbp1*	13	ENSMUSG00000032745	Comp (111562214..111626645)	155.66	−2.94	0.00
*Maz*	7	ENSMUSG00000030678	Comp (126621302..126626209)	70.90	−2.95	0.00
*Nobox*	6	ENSMUSG00000029736	Comp (43280608..43286488)	111.11	−2.96	0.00
*Cpeb1*	7	ENSMUSG00000025586	Comp (80996774..81105213)	146.89	−2.99	0.00
*E2f1*	2	ENSMUSG00000027490	Comp (154401327..154411812)	777.45	−3.02	0.01
*Klf2*	8	ENSMUSG00000055148	73072877..73075500	40.10	−3.18	0.02
*E2f5*	3	ENSMUSG00000027552	14643701..14671369	215.06	−3.21	0.00
*Zbed3*	13	ENSMUSG00000041995	95460120..95474349	2120.46	−3.68	0.00
*Mbd3*	10	ENSMUSG00000035478	Comp (80228373..80235384)	51.39	−3.71	0.00
*Jund*	8	ENSMUSG00000071076	71151599..71153265	359.34	−3.79	0.00
*Zfp213*	17	ENSMUSG00000071256	Comp (23775741..23783212)	23.71	−3.82	0.01
*Egr1*	18	ENSMUSG00000038418	34992876..34998037	398.93	−3.91	0.00
*Mbd6*	10	ENSMUSG00000025409	Comp (127117825..127124887)	14.95	−3.96	0.02
*Nme2*	11	ENSMUSG00000020857	Comp (93840640..93847085)	142.47	−4.08	0.00
*Jun*	4	ENSMUSG00000052684	Comp (94937271..94940459)	285.22	−4.14	0.00
*Junb*	8	ENSMUSG00000052837	Comp (85701113..85705347)	176.68	−4.42	0.00
*Fos*	12	ENSMUSG00000021250	85520664..85524047	672.23	−4.51	0.00
*Bmyc*	2	ENSMUSG00000049086	25596751..25597733	91.04	−5.89	0.00
*Ybx2*	11	ENSMUSG00000018554	69826622..69832431	205.82	−6.59	0.00

## Data Availability

All RNA-Seq data have been submitted to the SRA (SUB14127580).

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
