# Peer review of "Loss of ERβ Disrupts Gene Regulation in Primordial and Primary Follicles"

_ijms, 2024, doi:10.3390/ijms25063202_

Round 1
Reviewer 1 Report
Comments and Suggestions for Authors
In this manuscrpt, the author tried to explore the effects of ERb during PFGA using a ERb knockout mouse model. The topic is interesting and the overall design is reasonable. However, the analysis is very priliminary, the present of the results and discussion is rough. The sequencing results from this work deserve a much better mining and I believe there are more crucial results to be uncovered and more meanful conclusions to be made.
Here are some detailed probles as follow,
1 “PDF”and“PdF” were both used in the manuscript. Please unify the abbreviations.
2 L 36, The author mentioned 2 classes of Pdfs and 2 waves, please clarify them.
L56,what is PFA ?
3 Fig. 1C, How did the qPCR analized? If relative expression was caculated, fold change level of PdF sample should be 1. Also, how's the P value here?
4 L119,133, et.al., what are these genes? provide full names. It seems that some of them are LncRNAs?
5 L 163,There shoud be another subtitle for this part. Also, It is confused as PRF is follicle while PFGA is a process. The auther should reconsider all these expressions throughout the manuscript.
6 Can follicles in ERnull mouse PFGA? Is this PFGA physiologically the same as normal PFGA?
7 L187, Why the authors only identify the epigenetic regulators and TFs? Why not perform a KEGG analysis for all the possible regulative pathways?
8 Table 1-4 think most contents of these tables belongs to supplimentary material.
9 There are some layout problems in the manuscript, such as in L300.
Author Response
Overall comments
In this manuscript, the author tried to explore the effects of ERβ during PFGA using an ERß knockout mouse model. The topic is interesting, and the overall design is reasonable. However, the analysis is very preliminary, and the presentation of the results and discussion is rough. The sequencing results from this work deserve much better mining, and I believe there are more crucial results to be uncovered and more meaningful conclusions to be made.
Response # We are thankful to the reviewer for her/ her kind comments. We have addressed the individual comments in the following section.
Query #1 “PDF” and “PdF” were both used in the manuscript. Please unify the abbreviations.
Response #1 We have corrected the error and changed all PDF to PdF.
Query #2 L36: The author mentioned 2 classes of PdFs and 2 waves of PdFs; please clarify them.
Response #2 The two classes of PdFs develop in two waves. Therefore, the statements indicate precisely the same two groups of primordial follicles.
Query #3 L56, what is PFA?
Response #3 PFA represents Primordial Follicle Activation. It has been replaced by PFGA (Primordial Follicle Growth Activation) in the revised manuscript.
Query #4 Fig. 1C: How was the qPCR analyzed? If the relative expression was calculated, the fold change level of the PdF sample should be 1. Also, how's the P value here?
Response #4 The lowest value among the PdF (primordial follicle) group samples was considered 1. The rest of the PdF and PrF sample values were compared to the sample with the lowest expression (value 1). The p-value was > 0.05, which is indicated in our revised manuscript.
Query #5 L119,133, et.al., what are these genes? provide full names. It seems that some of them are LncRNAs?
Response #5 According to the reviewer’s suggestion, we have included two new tables of those genes indicating their full names and functional roles (Supplementary Table 1 and Supplementary Table 2 in the revised manuscript). KEGG analyses did not show any LncRNAs.
Query #6 L163: There should be another subtitle for this part. Also, it is confusing as PRF is a follicle while PFGA is a process. The authors should reconsider all these expressions throughout the manuscript.
Response #6 We are sorry for the confusion. The paragraph starting in L163 basically compares the genes stated in the above two paragraphs. If we add a new subtitle, that might indicate we are describing a different topic. But practically, the same topic is described throughout the four paragraphs under Section 2.4. Differential expression of follicular genes during PFGA.
Query #7 Can follicles in ERβnull mice undergo PFGA? Is this PFGA physiologically the same as normal PFGA?
Response #7 Yes. The primordial follicles in Erβnull mice undergo PFGA. However, the rate of PFGA is accelerated, there is an increased atresia of the activated follicles, and the activated follicles do not mature to ovulate, which has been reported previously. Therefore, physiologically, the PFGA in Erβnull mice cannot be considered normal.
Query #8 L187: Why do the authors only identify the epigenetic regulators and TFs? Why not perform a KEGG analysis for all the possible regulative pathways?
Response #8 Previous studies on Erβnull mice ovaries have identified genes related to steroidogenesis, preovulatory follicle maturation, and ovulation induction. In this study, we have emphasized whether the loss of ERβ impacts epigenetic and transcriptional regulators in ovarian follicles. KEGG analyses did not detect any LncRNAs, or any gene pathways related to ovarian dysfunction.
Query #9 Table 1-4: I think most of the contents of these tables belong to supplementary material.
Response #9 We agree with the reviewer that the list of genes should be shown in supplementary tables. We have shown the list of overall genes in the supplementary tables (Supplementary Tables 1 and 2). However, the epigenetic and transcription factors in primordial and primary follicles were not investigated previously. We, therefore, have included concise tables of those factors in the main body of the text section.
Query #10 There are some layout problems in the manuscript, such as in L300.
Response #10 We have corrected the layouts.
Reviewer 2 Report
Comments and Suggestions for Authors
Manuscript ID: ijms-2836324
Title: Loss of ERβ disrupts gene regulation in primordial and primary follicles
In this work, the authors had a particular interest in elucidating the role of Erb in the activation of primordial follicles. By a transcriptome analysis, they found an extensive panorama of the action of ERb over the expression of specific genes. I consider this to be very complete work. Undoubtedly, the results obtained respond to questions about the action of this transcription factor and will help to understand the mechanism of reactivation of the cell cycle in the oocyte.
Using Erb knockout mouse ovaries, the authors performed a transcriptome analysis with primordial and primary follicles, compared with control Erbfl/fl mouse ovaries. Their results showed a differential expression gene patron. They found genes that have a functional action over the activation of the primordial follicles. One of the genes that captured my attention was Dnmt1, which was downregulated because this gene is implicated in the methylation process.
The document followed an adequate scientific format. The introduction captures the reader's attention because it explains the importance of Erb in activating the primordial follicles very well and the implications of this premature activation. The authors cited all the affirmations with adequate references. The authors explore the possibility that the activation of primordial follicles is mediated by the action of Erb. To answer this question, the authors employ a suitable methodology. The results showed sufficient evidence of the action of Erb in the activation of primordial follicles. And these are correctly discussed.
Author Response
Overall comments of the Reviewer #2
Overall comments:
In this work, the authors had a particular interest in elucidating the role of ERβ in the activation of primordial follicles. By a transcriptome analysis, they found an extensive panorama of the action of ERβ over the expression of specific genes. I consider this to be very complete work. Undoubtedly, the results obtained respond to questions about the action of this transcription factor and will help to understand the mechanism of reactivation of the cell cycle in the oocyte.
Using ERβ knockout mouse ovaries, the authors performed a transcriptome analysis with primordial and primary follicles, compared with control Erβfl/fl mouse ovaries. Their results showed a differential expression gene patron. They found genes that have a functional action over the activation of the primordial follicles. One of the genes that captured my attention was Dnmt1, which was downregulated because this gene is implicated in the methylation process.
The document followed an adequate scientific format. The introduction captures the reader's attention because it explains the importance of ERβ in activating the primordial follicles well and the implications of this premature activation. The authors cited all the affirmations with adequate references. The authors explore the possibility that the activation of primordial follicles is mediated by the action of ERβ. To answer this question, the authors employ a suitable methodology. The results showed sufficient evidence of the action of ERβ in the activation of primordial follicles. And these are correctly discussed.
Our response# We are thankful to the reviewer for the kind comments.
Reviewer 3 Report
Comments and Suggestions for Authors
The manuscript is interesting, but before publication, I suggest some revisions. In the introduction, there is a lack of a clearly formulated research hypothesis. In the Materials and Methods section, it is unclear how many individuals were used for the isolation of vesicles, and there is no information on the viability of vesicles following enzymatic isolation. Have you checked it after completing the procedure? Additionally, there is a lack of statistical analysis - how could this oversight occur?
Author Response
Overall comments of the Reviewer #3
Query #1 In the introduction, there is a lack of a clearly formulated research hypothesis.
Response # 1. We have revised our research hypothesis in the last paragraph of the Introduction section.
Query #2. In the Materials and Methods section, it is unclear how many individuals were used for the isolation of follicles, and there is no information on the viability of follicles following enzymatic isolation. Have you checked it after completing the procedure?
Response # 2. The details of follicle isolation have been mentioned in our previous publication (Chakravarthi et al. 2020; PMID 32141511). Two individual researchers (V.P.C. and M.A.R.) were involved in the follicle isolation procedure. After enzymatic digestion and size fractionation, primordial and primary follicles were picked up under microscopic visualization. We only collected the follicles that were intact and morphologically appeared healthy. On the other hand, any follicles that appeared morphologically abnormal were excluded from the study. The enzymatic process used liberase, which gives highest number of viable follicles compared to other enzymatic digestion (Skulska et al, 2019; PMID: 31525366). Notably, when we performed follicle counting in whole ovary sectioning, atresia among the primordial and primary follicles was less frequent compared to those among larger follicles (Chakravarthi et al. 2020; PMID 32141511). We have shown the images of isolated follicles in Figure 1 and Figure 2.
Query # 3. Additionally, there is a lack of statistical analysis - how could this oversight occur?
Response # 3. We are very sorry for the mistake. We have included a Statistical Analysis (Section 4.7) in the revised manuscript.
Round 2
Reviewer 1 Report
Comments and Suggestions for Authors
The author have reasonably improved the manuscript.